# Tracing the History of Hepatitis E Virus Infection in Mexico: From the Enigmatic Genotype 2 to the Current Disease Situation

**DOI:** 10.3390/v15091911

**Published:** 2023-09-12

**Authors:** Oliver Viera-Segura, Arturo Calderón-Flores, Julio A. Batún-Alfaro, Nora A. Fierro

**Affiliations:** 1Laboratorio de Diagnóstico de Enfermedades Emergentes y Reemergentes, Centro Universitario de Ciencias de la Salud, Universidad de Guadalajara, Guadalajara 44340, Mexico; 2Departamento de Inmunología, Instituto de Investigaciones Biomédicas, Universidad Nacional Autónoma de México, Mexico City 04510, Mexico

**Keywords:** hepatitis E virus, viral hepatitis, HEV genotypes

## Abstract

Hepatitis E virus (HEV) is the major cause of acute viral hepatitis worldwide. This virus is responsible for waterborne outbreaks in low-income countries and zoonosis transmission in industrialized regions. Initially, considered self-limiting, HEV may also lead to chronic disease, and evidence supports that infection can be considered a systemic disease. In the late 1980s, Mexico became a hot spot in the study of HEV due to one of the first virus outbreaks in Latin America related to enterically transmitted viral non-A, non-B hepatitis. Viral stool particles recovered from Mexican viral hepatitis outbreaks represented the first identification of HEV genotype (Gt) 2 (Gt2) in the world. No new findings of HEV-Gt2 have been reported in the country, whereas this genotype has been found in countries on the African continent. Recent investigations in Mexico have identified other strains (HEV-Gt1 and -Gt3) and a high frequency of anti-HEV antibodies in animal and human populations. Herein, the potential reasons for the disappearance of HEV-Gt2 in Mexico and the advances in the study of HEV in the country are discussed along with challenges in studying this neglected pathogen. These pieces of information are expected to contribute to disease control in the entire Latin American region.

## 1. Introduction

Since hepatitis E virus (HEV) was identified 40 years ago as an etiological agent of self-limited hepatitis, its study has advanced rapidly in recent years [1]. This virus is currently considered the most common cause of acute hepatitis worldwide; it can lead to severe illness in pregnant women and patients with liver disease and chronic infections in immunosuppressed individuals [2]. Notably, its recently recognized ability to infect multiple cell types and the adverse effects observed in diverse tissues due to infection support the notion that hepatitis E might be considered a systemic disease rather than solely a liver disease [3,4,5,6,7,8,9].

HEV is a 7.2 kb positive single-stranded RNA virus that encodes from 3 to 4 open reading frames (ORFs): ORF-1 encodes nonstructural proteins required for virus replication [10], and ORF-2 and ORF-3 overlap and encode a protein that forms the viral capsid and a protein important for viral egress, respectively; an extra ORF-4 protein promotes virus replication [11]. The virus belongs to the *Paslahepevirus* genus within the *Hepeviridae* family. The *Paslahepevirus balayani* species comprises eight genotypes (Gts), of which five affect humans. HEV-Gt1 and HEV-Gt2 are restricted to infecting humans, while HEV-Gt3 and HEV-Gt4 infect both humans and diverse animal species, with pigs being the main reservoir with demonstrated zoonosis [12,13]. HEV-Gt7 infects camels and can also infect humans [14]. Zoonosis via rodent HEV belonging to the *Rocahepevirus ratti* species has recently been demonstrated [15].

The success of the global distribution of HEV has been attributed to its capacity to spread by several mechanisms: enteric transmission is frequent for HEV-Gt1 and HEV-Gt2 in developing regions, whereas transmission of HEV-Gt3 and HEV-Gt4 is commonly zoonotic in developed countries. All genotypes can be transmitted via the fecal-oral route through contaminated water and food, from mother to fetus, from person to person, and through blood derivatives [2,16]. Although several HEV genotypes have been identified with the capacity to infect humans, the serotype characterization of HEV is commonly focused on capsid protein reactivity, and only one serotype has been described, which allows the serological characterization of the HEV distribution in different regions [17,18].

Since the initial HEV finding in large waterborne outbreaks in low-income countries [19], the infection has largely been considered a poverty predictor; however, hepatitis E is still neglected in those countries. An example is the situation in the Latin American region, where the countries are developing regions and their inner socioeconomic structure is complex due to their economic inequity: many human realities are found in the subcontinent, from very rich areas in the cities and low-income, densely populated areas in the cities to very poor rural communities lacking modern sanitary services [20].

People living in Latin America are exposed to different HEV transmission and infection factors. Indeed, starting in the last century, Mexico has become an important country in the study of the viral dispersal and epidemiology of HEV due to one of the first recognized enterically transmitted viral non-A, non-B (NANB) outbreaks in Latin America in communities located in central Mexico. The viral stool particles recovered during those outbreaks, together with isolates from Asia [21,22,23], helped to describe and classify the molecular characteristics of the recently named hepatitis E virus [24]. This finding represented a milestone in the understanding of hepatitis viruses that infect humans and the first identification of HEV genotype (Gt) 2 (Gt2) in the world. Nonetheless, despite the relevance of the previously mentioned studies and even the identification of HEV-Gt3 in sera/fecal samples from farm swine in Mexico in the early 2000s, the genomic diversity of the virus was not revisited in the country until recently, when HEV-Gt1 and HEV-Gt3 were identified in human samples [25,26,27,28,29]. Furthermore, independent research groups in Mexico found a high proportion of individuals positive for anti-HEV antibodies.

The identification of three HEV Gts in Mexico, HEV-Gt1, HEV-Gt2, and HEV-Gt3, places the country as unique in America and denotes the need to continue with detailed epidemiological studies. This will allow us to dissect the role of hepatitis E in public health and will help resolve the multiple questions related to this virus, including an explanation for the apparent disappearance of HEV-Gt2 in Mexico. The revised information is relevant for the entire region since similar demographic and economic conditions prevail in most Latin American countries, which could predispose them to HEV transmission and infection.

## 2. HEV-Gt2: Its Emergence in Mexico

In the second half of 1986, two viral hepatitis outbreaks occurred in neighboring Hutzililla and Telixtac, rural villages in the state of Morelos, approximately 130 km south of Mexico City in the center of Mexico [30]. In the first outbreak at Hutzililla, 94 people showed an illness defined as jaundice (100%) accompanied by other symptoms, such as choluria (97.9%), anorexia (97.9%), asthenia (97.9%), and acholia (40.3%), during a period of 20 weeks in the rainy season starting in June [24,30]. The 94 affected subjects represented 5.4% of the population, with the highest disease prevalence found in the 15–24 age group. Sera samples from 62 of the affected individuals were analyzed for the presence of hepatitis A or B active infections. Nevertheless, no anti-HAV IgM, anti-HBV IgM antibodies, or HBV antigens were detected. Poor sanitary systems and contaminated water sources were the associated risk factors for the acquisition and spread of the disease; immune microscopy analysis of two collected stool samples from acutely infected patients showed viral particles of 32–34 nm. Among the infected patients, two nonpregnant adult women died, and one pregnant woman in the third trimester recovered from the disease; neither she nor her infant suffered any complications [30]. Data coincided with the enterically transmitted viral NANB hepatitis outbreaks in Asia, so the outbreak was described as one of the first NANB fecal-oral-transmitted hepatitis cases on the American continent [30].

In the second outbreak at Telixtac, 129 subjects, representing approximately 5% of the population, displayed the illness starting in August, also in the rainy season [30]. Two of 58 analyzed sera samples showed anti-HAV IgM antibodies, and one stool sample was positive for viral particles from 32 to 34 nm. Water contamination and contact with infected people were important and independent factors for illness acquisition in 64% of the cases; many of the male inhabitants of the two villages worked and ate their meals together at surrounding agricultural fields. In this second event, a nonpregnant adult woman died, and a pregnant woman delivered a 32-week premature infant who died of unknown causes at the age of three months. Demographic information indicated that the local farmers and their families did not have a history of recent travel outside of Mexico but that the area employed migrant workers from other states [30]. A follow-up of the patients did not show liver dysfunction over one year after the infection [30]. A third outbreak involving 38 subjects was registered a year later in a village two miles from Telixtac [30].

Viral particles isolated from Mexican viral hepatitis outbreaks were used for experimental infection of nonhuman primates. Some infected animals acquired hepatitis, showing that the virus caused the disease [22]. Additionally, cDNA obtained from viral RNA was expressed in lambda gt11 libraries; serological studies of the expressed proteins did show specific reactions with sera from patients from other enterically transmitted NANB viral hepatitis outbreaks [31], adding data to the growing evidence that the outbreaks in Mexico and those outbreaks in Asia were caused by the same or a closely related virus [21,32,33]. By the start of the 1990s, it was clear that the etiologic agent of NANB hepatitis was a new virus [33], and by 1991, the first complete sequence of the hepatitis E “Burma” strain obtained from samples from Burma was published [34].

Molecular characterization of the sequence obtained from the viral particles isolated from Mexican patients showed that they were homologous to the “Burma” HEV strain, with the same genome length and open reading frame organization, so the particles were ascribed to HEV [24]. The “Burma”, “Mexico”, and “Pakistan” strains were the only available HEV sequences at the time [24,30,33,35]. Burma and Pakistan were more similar, and the Mexican strain showed 76 and 77% identity matches with the Burma and Pakistan strains, respectively, in the overall nucleotide sequence [24]. Regarding the deduced amino acid sequence, the Mexican strain showed the lowest identity match in the ORF-1 region (13%) and the highest identity match in ORF-2, with a 93% identity match to the Burma and Pakistan strains [24]. The original isolate strain from Mexico was classified as HEV-Gt2 [36,37] and considered the HEV-Gt2 reference strain for the genotype [38] until another sequence of HEV-Gt 2 obtained from stored stool samples collected in Telixtac, Mexico, was published and became the reference sequence for the HEV-Gt2a GenBank KX571787 [39,40].

In addition to causing the 1986 outbreaks in Mexico, HEV-Gt2 has been found to be an agent of viral hepatitis in countries on the African continent, including Chad, Namibia, Nigeria, Senegal, and Sudan [41,42,43,44,45,46,47,48], mostly as outbreak episodes. HEV-Gt1 has also been found in hepatitis outbreaks in Africa alone or accompanied by HEV-Gt2 [42,44]. Hepatitis E outbreak events in African countries are related to deficient sanitary conditions and, in many cases, to very low-income human realities such as migration, mining work, and refugee camps [44]. The disease usually presents with jaundice, fever, vomiting, anorexia, and headache. Nevertheless, as noted by Desai et al., mild and asymptomatic cases may be unreported, as in sub-Saharan Africa, there are other causes of jaundice, such as malaria and yellow fever [42]. Molecular characterization of HEV-Gt2 is still very limited, as only three complete genomes, two from Mexico and one from Nigeria, have been published; six almost complete genomes are also available in GenBank. The only complete genome of HEV-Gt2 from Africa is the reference strain of genotype 2b [24,39,40,47].

## 3. The Enigmatic Disappearance of HEV-Gt2 in Mexico

Mexico is considered in the specialized literature to be an endemic country for hepatitis E; however, after the outbreaks in 1986, no cases of HEV-Gt2 have been reported in the country. The lack of recent evidence of HEV-Gt2 in Mexico is open to speculation. The diagnosis of viral hepatitis is well developed for the A, B, and C viruses in the country; since hepatitis E is considered “other” hepatitis, no routine diagnostics are run in the health system, and there are no specific epidemiologic data from the Ministry of Health [49]. Therefore, our current information comes from studies by independent research groups. From these studies, coinfection of hepatitis A virus (HAV) and HEV in poor children in Mexico has been reported [27]. Considering that HAV infection prevails in low-income regions and that diagnosis of HEV is clinically difficult since acute hepatitis E is often indistinguishable from liver disease caused by HAV, the contribution of HEV to the clinical manifestations of HAV-caused illness might be underreported, and circulation of HEV-Gt2 might therefore be underestimated.

Additionally, data from HEV-Gt2-associated disease come from symptomatic patients, and this genotype in the context of subclinical disease has not been studied in detail. Given the lack of surveillance and molecular epidemiology studies in open populations in Mexico [49], it is still plausible that HEV-Gt2 is circulating in the country. Sporadic cases may not be detected and likely contribute to the HEV reservoir responsible for maintaining the disease in a given population. Indeed, fecal shedding of the virus by individuals with subclinical infections could lead to the continuous maintenance of a source of infection in Mexico. In this regard, and even if water contamination is recognized as the main source of HEV-Gt2, the study of the virus in water remains unknown in Mexico.

Sanitary conditions in Mexico have been slowly improving, and HEV-Gt2, which appears mostly in very precarious sanitary conditions, may be less likely to be transmitted. However, thirty-seven years after the hepatitis E outbreaks in the country, a comprehensive follow-up serological and molecular study of the virus among the people affected in central Mexico by HEV-Gt2 is still lacking.

Finally, no animal reservoirs have been identified for HEV-Gt2. Thus, it appears unlikely that zoonotic transmission is responsible for infection with this genotype. Genotype selection is plausible during the evolution of the virus, and zoonotic HEV-Gt3 might be more efficiently spread, explaining the null detection of HEV-Gt2 in recent years in Mexico.

## 4. HEV Seroprevalence in Mexico

Different research groups have been interested in determining the seroepidemiology status of HEV in Mexico; an initial study reporting no anti-HEV antibody detection was published in 1994 [50]. The initial report was followed by a study of anti-HEV prevalence in 330 individuals from Veracruz and Mexico City, representing southeastern and central Mexico, respectively; the global reported frequency of anti-HEV antibodies was 3.9%, with the highest contribution attributed to subjects from rural areas of Veracruz (6.8%) [51]. Additionally, the evaluation of anti-HEV antibody presence in sera samples from 363 volunteers from Hidalgo, located in the center of Mexico, showed a seroprevalence of 6.3%. Being men older than 50 years and/or living in poor socioeconomic conditions were more frequently associated with predisposition factors for anti-HEV positivity [52].

After the initial reports underscoring HEV circulation in Mexico, national anti-HEV seroprevalence was evaluated in 1999 by the analysis of 3549 samples from subjects aged from 1 to 29 years old from all 32 states in the country; the authors reported that 10.5% of the studied individuals did have anti-HEV antibodies [53] Quintana Roo, located in southern Mexico, showed the highest prevalence (n = 512, 23%), followed by Chiapas, Guerrero, and Oaxaca, also located in southern Mexico (n = 194, 14.9%). The age group with the lowest proportion of participants with anti-HEV antibodies was the 1–4 years group; an increasing percentage of anti-HEV positivity as age increased was found, reaching the maximum incidence among people aged 26–29 years. The main risk factors identified for HEV infection were age, educational level, and living in rural communities [53]. Another evaluation of anti-HEV prevalence was conducted in 2016 with 624 samples from individuals from all the states in Mexico. The reported anti-HEV prevalence was 1.76%, and the main contributions were derived from the states of Baja California, Colima, and Campeche, located in the north, west, and south, respectively [54].

The information available for the northern state of Durango showed 36.5 and 40.7% (n = 146 and n = 150 respectively) anti-HEV prevalence in subjects from two rural communities but a lower prevalence (4.5%, n = 425) in the urban area of Durango City [55]. Two ethnic groups with settlements in rural areas of the state were also evaluated, showing a prevalence of 6.7% (n = 150) and 3.4% (n = 146) among Mennonites and Tepehuanos, respectively [56,57]. The fact that the rural population did show higher anti-HEV seroprevalence compared to the urban and ethnic populations has been previously attributed to the poor sanitation in rural communities compared to other communities [58].

Mexico is highly endemic for HAV; therefore, this infection is common in the pediatric population [59]. This situation is a bias for diagnosing HEV in pediatric patients with acute hepatitis. A serological evaluation of pediatric patients with acute hepatitis from the state of Jalisco located in western Mexico demonstrated that 50% of the children did have a coinfection of HAV and HEV, while 3% were infected only with HEV (n = 206) [26]. Another study analyzed sera from pediatric patients with acute hepatitis from the states of Jalisco and Oaxaca (western and southern Mexico, respectively). The detection of antibodies against HAV and HEV in Jalisco and Oaxaca with 129 and 93 pediatricians found no mono-infection for HEV, with a 58% and 10% seroprevalence of anti-HEV IgM as coinfection, respectively, demonstrating a very recent or active infection [60]. Moreover, in the center of Mexico, in Mexico City and Mexico State, 3% and 6% of anti-HEV IgG and IgM antibodies, respectively, were reported [29]. Therefore, differences in the distribution of HEV exist and may be related to distinct sources of the virus. Importantly, since there is not yet a gold standard for the detection of anti-HEV antibodies, distinct assays have been used in Mexico. These include kits manufactured in the USA, Germany, China, and Switzerland, all of which have differences in the limit of detection. Therefore, the results obtained might diverge. This must be considered when comparisons between distinct studies are conducted.

It is important to point out that, as for other infectious etiologies, the picture of hepatitis E in Mexico may affect the virus’s global distribution. In this sense, one of the countries in the Americas with the highest HEV seroprevalence is the USA, with a mean of 9% [61]. The USA and Mexico share a high seroprevalence of HEV and one of the largest borders in the world [62], with a very active flow of people in both directions. In this context, several case reports of individuals who traveled to Mexico and developed hepatitis E [63,64,65,66,67,68] support the necessity of adequate measures to follow infection in Mexico to avoid virus dissemination.

## 5. HEV in Risk Groups

Specific human groups are known to have higher HEV infection risk; anti-HEV seroprevalence in those groups has been evaluated in Mexico. Pregnant women are a high-risk group for HEV infection due to the probability of developing severe acute liver disease and a high percentage of fatal outcomes during pregnancy, particularly during the third trimester [69]. Two independent studies conducted in Durango and Mexico City showed anti-HEV antibody distributions of 5.7% and 7.4% (n = 439 and n = 428 respectively), respectively. From these studies, the presence of antibodies was mostly associated with the consumption of unpasteurized cow milk [70,71]. Evaluation of anti-HEV antibodies in low-income pregnant women in El Paso-Ciudad Juarez (USA and Mexico border) showed 0.4 and 1.6% positivity, respectively [72]. Another report on the evaluation of anti-HEV IgG in pregnant women was conducted in a group of medium to high socioeconomic patients of a hospital in Mexico City, and the prevalence was 0.79% (n = 127); the lower antibody prevalence compared to other groups was explained as being possibly connected to socioeconomic status [71].

People with liver transplants are another high-risk group for HEV infection, as some HEV genotypes might progress to chronic infection and then liver damage during immunosuppression. A group of 200 liver transplantation recipients from Mexico City was evaluated for anti-HEV presence three or more months after the intervention; the study revealed that the patients had been in contact with HEV, as the IgG prevalence was 15% and the IgM prevalence was 5% [73].

Waste pickers and butchers can be considered in occupational exposure to HEV, as they work with human sanitary residues or animal products. In a serological survey conducted in the state of Durango, an anti-HEV IgG prevalence of 16.3% was found among the waste pickers group [74], and a prevalence of 17.8% was found for the group of butchers [75]. In both groups, the seroprevalence was higher than the seroprevalence of the control group. IgG detection showed exposure to HEV in the past, but anti-HEV IgM presence was not evaluated in the analyzed groups. Therefore, a current HEV infection at that time could have been missed [74,75].

The development of a chronic HEV infection has been linked to immunosuppression, and HEV could lead to severe illness in patients with chronic liver disease [76]. A study that aimed to analyze possible HEV infections in patients with chronic liver damage in Mexico found that in the state of Jalisco, 29.4% of a group of 513 liver disease patients were positive for anti-HEV antibodies. Interestingly, when patients were categorized excluding other infectious comorbidities (HIV and Hepatitis A, B, and C) and grouped according to high alcohol consumption and obesity, obese patients were the ones with the highest frequency of antibodies (57.5%), providing the first evidence of a relation between obesity and hepatitis E [77]. The high frequency of antibodies in patients with liver disease is consistent with 26% of antibodies previously reported in cirrhotic patients in the same region [78] and highlights the important role of this virus in affecting health in the country.

## 6. Potential Sources of HEV in Mexico

Altogether, these results illustrate the different frequencies in the circulation of HEV throughout the country, probably due to the distinct realities in Mexico and to the practices immeasurable in cultural aspects. This impacts the transmission routes of the virus (Figure 1). In this regard, sera from 691 blood donors from the state of Jalisco were evaluated for anti-HEV presence. The reported frequency was 9.4% of blood donors reactive to anti-HEV IgG and 7.1% reactive to both IgG and IgM. Reactive sera were pooled, and HEV RNA was found at an approximate rate of 0.14% [79]. This rate of viremic donors is comparable to that reported in highly endemic regions in Asia [80] and supports that this virus is circulating widely in the country. The fact that blood donors did show very recent or active HEV infection emphasizes the need for HEV screening in Mexican blood banks to avoid the potential for virus transmission.

The zoonotic HEV transmission route has been proven for genotypes 3, 4, and 7, and wild animal populations, as well as domestic or livestock animals, can acquire HEV infection, passing the virus to humans and/or acting as viral reservoirs, representing a risk for humans in contact with the animals and their products and for people eating raw or insufficiently cooked meat of infected animals [81]. In this regard, monitoring wild, domestic, and livestock animals is crucial. An analysis conducted in 2012 evaluated the presence of anti-HEV antibodies in wild white-tailed deer of northern Mexico and revealed a 62.7% (n = 347) anti-HEV IgG prevalence [82]. Regarding domestic pigs, several studies have been performed with alarming results; in Durango, an incidence of 34.8% anti-HEV IgG antibodies was observed in backyard pigs, but the anti-HEV prevalence in slaughtered pigs intended for human consumption in Durango City was 79.8% [83]. This was in line with the high seroprevalence reported by Merino-Ramos T. et al. and García-Hernandez M., whose studies indicated a high incidence of anti-HEV antibodies in farm pigs that reached 90% in Jalisco state and 86.6% on farms in northern Mexico [84,85]. These results are crucial for the health and production sectors because these animals are intended for human consumption and represent an underestimated source of HEV infection. Mexico is classified as the eighth largest producer of pigs for consumption [86]. Therefore, to cut the chain of transmission, characterizing genomic diversity in recognized sources is a priority.

## 7. The Molecular Epidemiology of HEV in Mexico: A Challenge to Work on

Despite the relevance of HEV-Gt2 findings in the late 1980s, no molecular investigations on human HEV infections were performed in Mexico until 2018, when the circulation of HEV-Gt1 was confirmed by the sequencing of two genome regions corresponding to ORF-2 (nucleotides 6007–6303) and ORF-2/3 (nucleotides 6468–7113) in samples from pediatric patients with acute hepatitis [27]. Following the same strategy, HEV-Gt3 was reported from the retrospective analysis of samples from patients with chronic liver disease in 2020 [28].

The study of animal reservoirs in the country did not receive attention even though swine were identified as a reservoir of HEV at the end of the 1990s [87], adequate control policies were not implemented, and studies were not performed to evaluate the circulation of this etiological agent in production animals. In 2005, Cooper et al. performed a molecular study using sera/fecal samples from farm swine in northern Mexico. The authors reported the identification of HEV-Gt3 by sequencing, representing the first report of this genotype in the country [25].

Later, in 2013, a study was carried out using livers from pigs in northern Nuevo Leon state intended for human consumption. The author obtained a partial genome corresponding to ORF-2 from nine liver samples; consecutive sequencing analysis showed that the genome corresponded to HEV-Gt3 [88]. This molecular epidemiological description of the circulation of HEV-Gt3 in farm pigs from Mexico was complemented by a study carried out in samples from the center of Mexico where the circulation of HEV-Gt3 on swine was confirmed over the complete genome sequencing of one sample and two partial genomes [89]. In association with serological evaluation, these studies confirmed the endemicity of HEV in animals intended for human consumption, emphasizing the need for adequate food safety and surveillance systems.

Differences in the distribution of genotypes throughout the country have been found. In humans, HEV-Gt1 has been found in the west of the country, HEV-Gt2a in the center of the country, and HEV-Gt3 in the west and center of the country. In swine, HEV-Gt3 has been reported in the north, west, and center of the country, while HEV-Gt3a has been found in the center of the country (Figure 2). The finding of three genotypes in Mexico places the country as unique in Latin America, where HEV-Gt1 and -Gt3 have been found in humans and HEV-Gt3 in swine (Figure 2).

As can be inferred from the information above, there was a gap in HEV molecular detection in Mexico between the HEV-Gt2 outbreaks of the 1980s and HEV RNA detection in recent years. Molecular detection of HEV-RNA in clinical and epidemiological contexts pursues two main goals: diagnosis of active infections and genome sequencing for genotyping. In Mexico, only scientific studies have been published addressing molecular HEV detection, and for most Latin American countries, short partial sequences of the HEV genome are commonly reported (Appendix A) [25,27,28,29,40,89,93,94,95,96,97,98,99,100,101,102,103,104,105,106,107,108,109,110,111,112,113,114,115,116,117,118,119,120,121,122,123,124,125,126,127,128,129,130,131,132,133,134,135,136,137,138,139].

The fact that there is no routine HEV molecular detection in the Mexican health system implies that most of the acute infections are missed, as well as the opportunity to obtain information regarding HEV genotype circulation in time. Some authors have reported working with old sera samples, making it difficult to obtain epidemiological data across time. From the HEV partial sequences obtained in Mexico and reported to GenBank, phylogenetic trees of ORF1, ORF2, and ORF3 alignments were built separately, as shown in Figure 3. Most of the ORF1 sequences obtained in Mexico are from humans and form a branch of HEV-Gt3. Additionally, they are closely related and differ from the ORF1 sequences obtained from swine in Mexico. As HEV-Gt3 does have subgenotypes, longer sequences are needed to assign subgenotypes. HEV ORF3 sequences obtained in Mexico are mostly from swine; they form a branch of closely related sequences, and as noted by Cooper et al., it is possible to recognize the geographic origin of the sequences as they are grouped [25]. It is interesting to note that there was a clade of sequences obtained in 2005 (AY858909, 33, 34, 35, and 36) and sequences deposited in GenBank in 2017 (MF116409, 10, 11, and 12); unfortunately, as the MF series of sequences do not have an associate publication, the geographic origin is not known. HEV ORF3 human sequences obtained in Mexico clearly belong to the HEV-Gt1 and HEV-Gt3 branches as well as the partial ORF2 sequences that belong to HEV-Gt1. From this analysis, we can conclude that a short HEV sequence seems to be sufficient for assigning genotypes, but especially for HEV-Gt3, which does have several subgenotypes, longer or complete HEV sequences are needed.

## 8. Conclusions

The burden and impact of HEV in Mexico are underestimated due to the lack of an effective system for monitoring this infection. This can be attributed to six reasons: (1) acute hepatitis E is clinically indistinguishable from other hepatitis viruses, and molecular HEV detection is not common; (2) HEV-related disease is mostly self-limited, and therefore, no follow-up is carried out for ill patients; (3) the lack of hepatitis surveillance in open populations prevents the identification of subclinical HEV infections, which in turn could contribute to virus underestimation and the spread of infection; (4) swine populations, water contamination, and blood derivatives might contribute to the circulation of HEV; however, there is a lack of surveillance on farms, and minimal studies regarding HEV detection in water and blood banks have been reported in Mexico; (5) the consequences of infection for specific risk groups (pregnancy, hepatopathies, and immunosuppression) have not been studied in detail in the country; and (6) very importantly, education of the population and health professionals on the transmission pathways of HEV and the appropriate diagnosis is not implemented in Mexico.

“He who seeks finds” is a popular Mexican saying; this applies to a large extent to the hepatitis E situation in the country, where we still lack a real picture of the infection that has been with us for at least thirty-seven years.

Despite the relevance of HEV in Latin America given the demographic and economic conditions that predispose the region to virus circulation, the available literature is still limited there. The pieces of information discussed here are expected to contribute to hepatitis E control in the entire Latin American region.

## Figures and Tables

**Figure 1 viruses-15-01911-f001:**
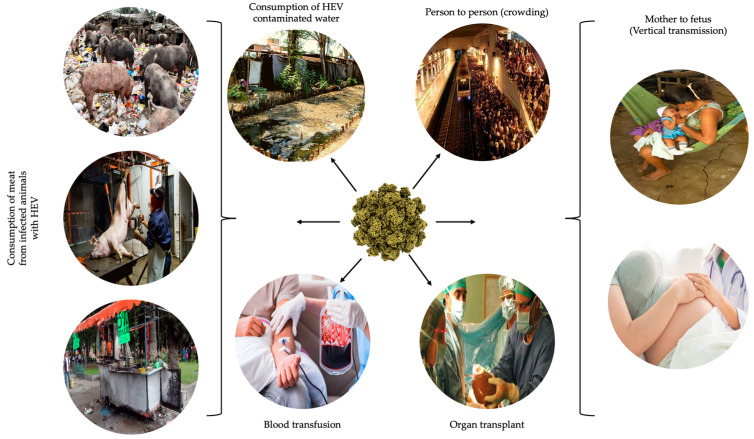
Potential sources of HEV in Mexico.

**Figure 2 viruses-15-01911-f002:**
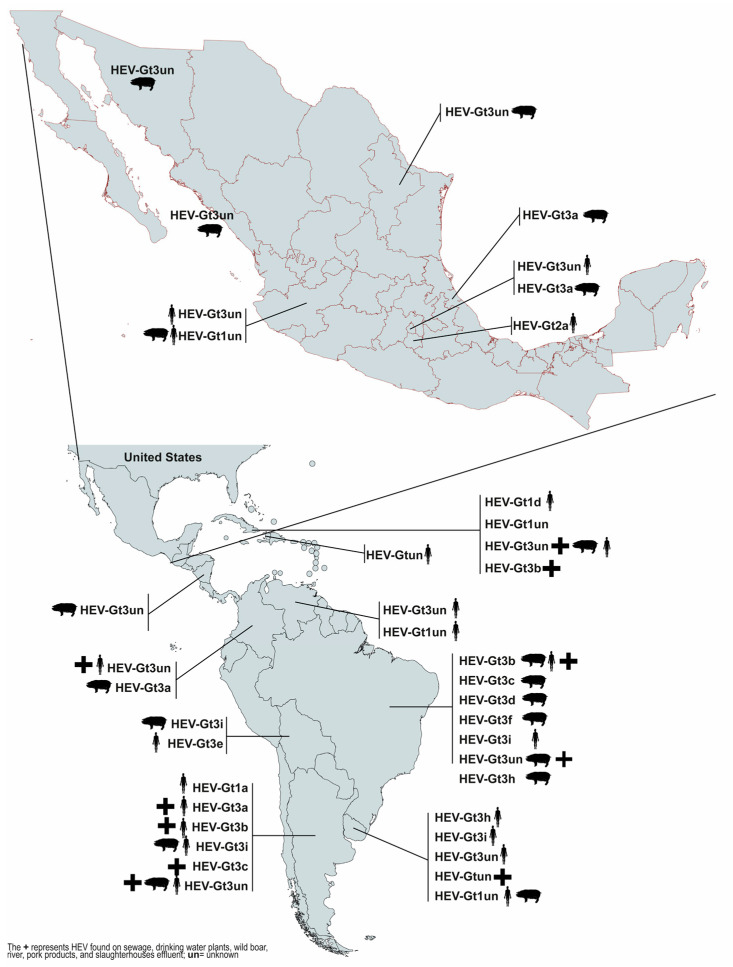
Distribution of hepatitis E virus in Latin America. The most prevalent host studies in the region are human and swine, but dolphins and deer have also been reported, together with sewage [90,91,92].

**Figure 3 viruses-15-01911-f003:**
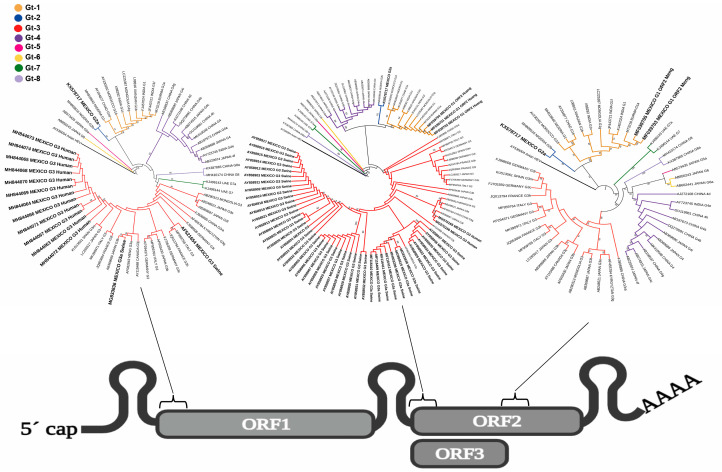
Phylogenetic tree of HEV partial genome sequences. The phylogenetic analysis was carried out by the maximum likelihood method based on the sequences from three HEV-genome regions, ORF1 (287 pb, from 56 to 343), the ORF-2/3 overlapping region (132 pb, from 5325 to 5457), and ORF-2 (297 pb, from 6007 to 6304 and 645 pb, from 6468 to 7113). Bootstrap values were determined with 10,000 resamplings of the datasets. Mexico sequences are highlighted in bold and in larger font sizes. The reference sequences are identified by GenBank accession number. A phylogenetic tree was constructed using RStudio software V.4.2.1 and the Itool web server.

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
