# Peer review of "Tracing the History of Hepatitis E Virus Infection in Mexico: From the Enigmatic Genotype 2 to the Current Disease Situation"

_viruses, 2023, doi:10.3390/v15091911_

Round 1

Reviewer 1 Report

This manuscript is a review on HEV infection in Mexico from the first identification of HEV genotype 2 in 1986 to the current situation indicating the circulation of 2 HEV genotypes (HEV-1 and HEV-3). However, recent data are very limited and the presence of HEV genotype 2 in some populations cannot be completely excluded.

This topic is interesting and the manuscript is well written.

Comments

1.       Abstract and text :

The uncommon detection worldwide of HEV-2 is mentionned. This is true for HEV-2a (Mexico strain) but not for HEV-2b (Nigeria strain) that is frequently detected in several countries in Africa

2.       Reference 11 is not adequate. ORF4 has been described by Nair in Plos Pathogens 2016

3.       HEV seroprevalence in Mexico

It is important to mention that the HEV seroprevalence is strongly influenced by the type of IgG assay used from one study to another due to differences in the limit of detection (expressed in WHO units/ml).

4.       Potential sources of HEV in Mexico

HEV RNA can be detected in negative anti-HEV blood donors in two third of cases

5.       Supplementary Table 1 : a column « Year of detection » should be added

This manuscript is a review on HEV infection in Mexico from the first identification of HEV genotype 2 in 1986 to the current situation indicating the circulation of 2 HEV genotypes (HEV-1 and HEV-3). However, recent data are very limited and the presence of HEV genotype 2 in some populations cannot be completely excluded.

This topic is interesting and the manuscript is well written.

Comments

1.       Abstract and text :

The uncommon detection worldwide of HEV-2 is mentionned. This is true for HEV-2a (Mexico strain) but not for HEV-2b (Nigeria strain) that is frequently detected in several countries in Africa

2.       Reference 11 is not adequate. ORF4 has been described by Nair in Plos Pathogens 2016

3.       HEV seroprevalence in Mexico

It is important to mention that the HEV seroprevalence is strongly influenced by the type of IgG assay used from one study to another due to differences in the limit of detection (expressed in WHO units/ml).

4.       Potential sources of HEV in Mexico

HEV RNA can be detected in negative anti-HEV blood donors in two third of cases

5.       Supplementary Table 1 : a column « Year of detection » should be added

Author Response

  1. Abstract and text: The uncommon detection worldwide of HEV-2 is mentioned. This is true for HEV-2a (Mexico strain) but not for HEV-2b (Nigeria strain), which is frequently detected in several countries in Africa.

As described by the reviewer, HEV-2 has been detected in several countries in Africa; therefore, its detection is not uncommon. The new version was amended accordingly.

  1. Reference 11 is not adequate. ORF4 was described by Nair in Plos Pathogens 2016.

We agree that the Nair et al. 2016 reference is more adequate. The previous reference was changed.

  1. HEV seroprevalence in Mexico: It is important to mention that the HEV seroprevalence is strongly influenced by the type of IgG assay used from one study to another due to differences in the limit of detection (expressed in WHO units/ml).

As pointed out by the reviewer, since there is not yet a gold standard for the detection of anti-HEV antibodies, distinct assays have been used in Mexico. These include kits manufactured in the USA, Germany, China and Switzerland, all of which have differences in the limit of detection. Therefore, the results obtained might diverge. This must be considered when comparisons between distinct studies are conducted. This information is included in the revised version.

  1. Potential sources of HEV in Mexico: HEV RNA can be detected in negative anti-HEV blood donors in two-thirds of cases

Thanks to the reviewer’s comment, we identified that the text required to be amended. Briefly, sera from blood donors from Mexico were evaluated for anti-HEV presence. The reported frequency was 9.4% of blood donors reactive to anti-HEV IgG and 7.1% reactive to both IgG and IgM. Reactive sera were pooled, and HEV RNA was found at an approximate rate of 0.14% (Copado-Villagrana 2023). This rate of viremic donors is comparable to that reported in highly endemic regions in Asia (Wolski A 2023) and supports that this virus is circulating widely in the country. The fact that blood donors did show very recent or active HEV infection emphasizes the need for HEV screening in Mexican blood banks to avoid the potential of virus transmission.

  1. Supplementary Table 1: a column « Year of detection » should be added

As recommended by the reviewer, the year of detection was added to the table.

The authors appreciate the critical analysis by the reviewer.

Reviewer 2 Report

This review article provides an exposition on the emergence and subsequent enigmatic disappearance of genotype 2 of Hepatitis E virus (HEV-Gt2), its seroprevalence, prevalence within risk groups, potential sources, molecular epidemiology in Mexico, and its distribution across Latin America. The manuscript is not only well-crafted and engaging but also imparts valuable insights to readers invested in the realm of HEV research.

There exist only minor concerns and suggestions that necessitate attention, as delineated below.

Comments:

1.         Minor editing of English language is required. For example, "Blood sera" should be changed to "sera" (Page 6, lines 287 and 290).

2.         Figure 1 is less informative and would benefit from removal.

3.         Figure 2: The sentence elucidating "+" is marred by typographical errors such as "segge," "effkluent," "board," and "unknow."

4.         Figure 3: Given the absence of a phylogenetic tree constructed from complete genomic sequences in this figure, the title warrants modification. Additionally, the phylogenetic trees appear excessively diminutive and indistinct, necessitating enlargement. Genotypes and corresponding bootstrap values ought to be duly marked on the trees. At the base of the figure, explicit details about the three genomic regions, along with their lengths and nucleotide positions (with reference to M73218), where the phylogenetic trees were formulated, should be incorporated.

5.         Supplementary Table 1: Each strain should be accompanied by the length of its genomic sequence.

Minor editing of English language is required

Author Response

The authors deeply appreciate the inspiring words to continue the effort in the study of this fascinating and enigmatic virus.

  1. Minor editing of English language is needed. For example, "Blood sera" should be changed to "sera" (Page 6, lines 287 and 290).

These changes have been made according to the reviewer’s comment.

  1. Figure 1 is less informative and would benefit from removal.

As pointed out by the reviewer the sources of HEV in Mexico are similar to those described worldwide. However, they are also influenced by characteristics related to practices immeasurable in cultural aspects of the country. Figure 1 was modified, and its purpose is to illustrate those peculiarities of Mexico that may also be related to the circulation of the virus in Latin America, since apart from the demographic and economic conditions that predispose the region to virus circulation, cultural aspects are also common in many ways.

  1. Figure 2: The sentence elucidating "+" is marred by typographical errors such as "segge," "effkluent," "board," and "unknow."

Typographical errors were corrected.

  1. Figure 3: Given the absence of a phylogenetic tree constructed from complete genomic sequences in this figure, the title warrants modification. Additionally, the phylogenetic trees appear excessively diminutive and indistinct, necessitating enlargement. Genotypes and corresponding bootstrap values ought to be duly marked on the trees. At the base of the figure, explicit details about the three genomic regions, along with their lengths and nucleotide positions (with reference to M73218), where the phylogenetic trees were formulated, should be incorporated.

As pointed out by the reviewer, Figure 3 legend was incorrect and has been modified in the revised version. Additionally, changes have been made according to the reviewer’s comment.

  1. Supplementary Table 1: Each strain should be accompanied by the length of its genomic sequence.

As recommended by the reviewer, the length of genomic sequences was added to the table.

  1. Minor editing of English language is needed.

The manuscript has been professionally reviewed by an editorial service and a native english speaker to better fit the language standards.

The authors appreciate the critical analysis by the reviewer.